# Enterovirus particles expel capsid pentamers to enable genome release

David Buchta [1], Tibor Füzik [1], Dominik Hrebík [1], Yevgen Levdansky[1,3], Lukáš Sukeník[1,2], Liya Mukhamedova[1], Jana Moravcová[1], Robert Vácha[1,2] & Pavel Plevka [1]

Viruses from the genus *Enterovirus* are important human pathogens. Receptor binding or exposure to acidic pH in endosomes converts enterovirus particles to an activated state that is required for genome release. However, the mechanism of enterovirus uncoating is not well understood. Here, we use cryo-electron microscopy to visualize virions of human echovirus 18 in the process of genome release. We discover that the exit of the RNA from the particle of echovirus 18 results in a loss of one, two, or three adjacent capsid-protein pentamers. The opening in the capsid, which is more than 120 Å in diameter, enables the release of the genome without the need to unwind its putative double-stranded RNA segments. We also detect capsids lacking pentamers during genome release from echovirus 30. Thus, our findings uncover a mechanism of enterovirus genome release that could become target for antiviral drugs.

[1] Central European Institute of Technology, Masaryk University, Kamenice 5, Brno 625 00, Czech Republic. [2] Faculty of Science, Masaryk University, Kamenice 5, Brno 625 00, Czech Republic. [3] Present address: Max Planck Institute for Developmental Biology, Max-Planck-Ring 5, 72076 Tübingen, Germany. Correspondence and requests for materials should be addressed to P.P. (email: pavel.plevka@ceitec.muni.cz)

Viruses from the family *Picornaviridae* form non-enveloped icosahedral virions that are about 30 nm in diameter. Picornavirus capsids protect 8000-nucleotide-long single-stranded RNA genomes, which are translated into polyproteins that are co-translationally and post-translationally cleaved by viral proteases to produce structural (capsid-forming) and non-structural proteins[1]. The capsid proteins VP1, VP2, VP3, and VP4 originating from a single polyprotein form a protomer, the basic building block of the icosahedral capsid. The entire capsid consists of 60 such protomers, arranged in 12 pentamer units.

The interactions of enteroviruses with receptors or exposure to acidic pH in endosomes induce conformational changes in virions into an activated state characterized by increased particle diameter, reduced contact areas between pentamers of capsid protein protomers, release of VP4 subunits from particles, and externalization of the N termini of VP1 subunits[1–5]. The activated particles of numerous enteroviruses were shown to contain openings along two-fold ($5 \times 10$ Å) or five-fold (diameters of up to 8 Å) axes of icosahedral symmetry of their capsids[2–6]. It has been speculated that these pores serve for the release of enterovirus genomes[3,6–10]. However, capsids of viruses from the families *Reoviridae* and *Totiviridae* that release single-stranded RNAs as part of their replication cycles contain circular pores larger than 15 Å in diameter[11,12]. The size of the pores in enterovirus capsids is not sufficient for the passage of single-stranded RNA[2–6,13–15]. Furthermore, enterovirus genomes contain sequences that form double-stranded RNA segments, which fold into three-dimensional (3D) structures, such as the internal ribosomal entry site required to initiate translation of viral RNA[16]. If these double-stranded RNA segments were present inside enterovirus particles, then the genome release would require either the opening of pores larger than 40 Å in diameter, or a mechanism to unwind the double-stranded RNA. However, there is no evidence of an association between enzymes with RNA helicase activity and enterovirus virions[1,17]. The structures of enterovirus particles before and after genome release have been characterized at high resolution[3,6–10]. Most of these cryo-electron microscopy (cryo-EM) and X-ray crystallography studies imposed icosahedral symmetry during the structure determination process and were not aimed at identifying the unique site of genome exit[3–7,9]. Asymmetric single-particle reconstruction and sub-tomogram averaging studies, at a resolution of 50 Å, were used to indicate that RNA exits poliovirus particles close to a two-fold axis[8]. The end stage of the enterovirus genome release are the empty capsids, the structures of which were determined for several enteroviruses[4,5].

Here we show that the exit of the RNA from the particles of echoviruses 18 and 30 requires capsid opening and results in a loss of up to three adjacent capsid protein pentamers. The large openings in the capsid enable the release of the genomes without uncoiling their double-stranded RNA segments.

## Results and Discussion

### Opening of particles enables genome release of echovirus 18.

We imaged enterovirus particles in the process of genome release by cryo-EM. Specifically, we performed cryo-EM of echovirus 18 virions exposed to pH 6.0 for 10 min, mimicking the acidic environment that the virus encounters in endosomes (Fig. 1a, Supplementary Fig. 1). Reference-free two-dimensional (2D) class averages show that the particles releasing genomes lack parts of their capsids (Fig. 1b). Asymmetric reconstruction combined with 3D classification identified subpopulations of echovirus 18 particles that lacked up to three pentamers of capsid protein protomers (Fig. 2a–c, Supplementary Fig. 2a). The missing pentamers always formed a single compact opening through the

capsid (Fig. 2a–c). We call the particles lacking one or several pentamers open particles. The remaining particles with complete capsids were either activated particles or empty capsids (Supplementary Fig. 2a). We did not detect native echovirus 18 virions that lacked pentamers at neutral pH. The asymmetric reconstructions of the open particles were determined to resolutions better than 9 Å (Supplementary Table 1, Supplementary Figs 3–5). The absence of one pentamer of capsid protein protomers creates a 120 Å-diameter pore in the capsid (Fig. 2a). The openings formed by the removal of one or more pentamers are sufficiently large to allow release of the viral RNA, even if the genome contains double-stranded RNA segments. Individual micrographs of the open particles frequently show multiple strands of RNA passing through the pore (Fig. 1b). In the 3D reconstructions, the capsid openings contain featureless electron densities with average values two times lower than those of the protein capsid (Fig. 2d–i). This diffuse electron density corresponds to an average of the RNA genomes escaping from the virions, which have unique conformations in each of the particles included in the reconstructions (Fig. 1b). In contrast to the formation of the open particles of echovirus 18, complete capsids were observed in the cryo-EM study of the genome release of poliovirus[8]. However, the poliovirus uncoating was induced by the exposure of particles to 56 °C, which may have affected the secondary structure of the genome and the mechanism of its release. A comparison of the distribution of charges and hydrophobic areas at the inter-pentamer contacts (Supplementary Fig. 6) and a comparison of the inter-pentamer buried surface areas (Supplementary Table 2) of enteroviruses do not provide evidence why distinct enteroviruses should employ different genome release mechanisms.

### High-resolution structures of open particles of echovirus 18.

Reconstructions of the open particles missing one, two, or three pentamers with imposed five-fold, two-fold, and three-fold symmetries, respectively, were determined to resolutions of 3.8, 4.1, and 3.7 Å, which allowed their molecular structures to be built (Fig. 3, Supplementary Fig. 4, Supplementary Table 1). Except for the missing pentamers, the open particles are similar in structure to that of the activated echovirus 18 particle: increased diameter relative to the native virus, reduced inter-pentamer contacts, absence of VP4 subunits, and holes along icosahedral two-fold symmetry axes (Fig. 3, Supplementary Fig. 7)[2–5,14,15]. The capsid proteins next to the missing pentamers are more mobile than the rest of the capsid, as indicated by the three times higher temperature factors than the remainder of the capsid. Residues 1–42 from the N termini of VP1 subunits, which were previously shown to interact with membranes and facilitate enterovirus genome delivery into the cytoplasm[1,18,19], are not resolved in the electron density map, indicating that their structures differ among particles. The N-termini of some of the VP1 subunits could reach out of the capsid through the openings formed by the missing pentamers. It has been shown that the RNA genome of poliovirus is protected against RNase A degradation during uncoating and transfer across the membrane[20]. It may seem that opening the particles of echovirus 18 exposes their genomes for degradation. Nevertheless, in silico simulations (Fig. 4, Supplementary Movie 1) and considerations of genome diffusion from the capsid (for details, see Methods) show that the large capsid opening results in a microsecond release time of the genome. The short time required for genome release limits the potential for its degradation. The externalized VP4 subunits and N termini of VP1 were shown to be required for the subsequent transport of enterovirus genomes across the endosome membranes[18,21–24].

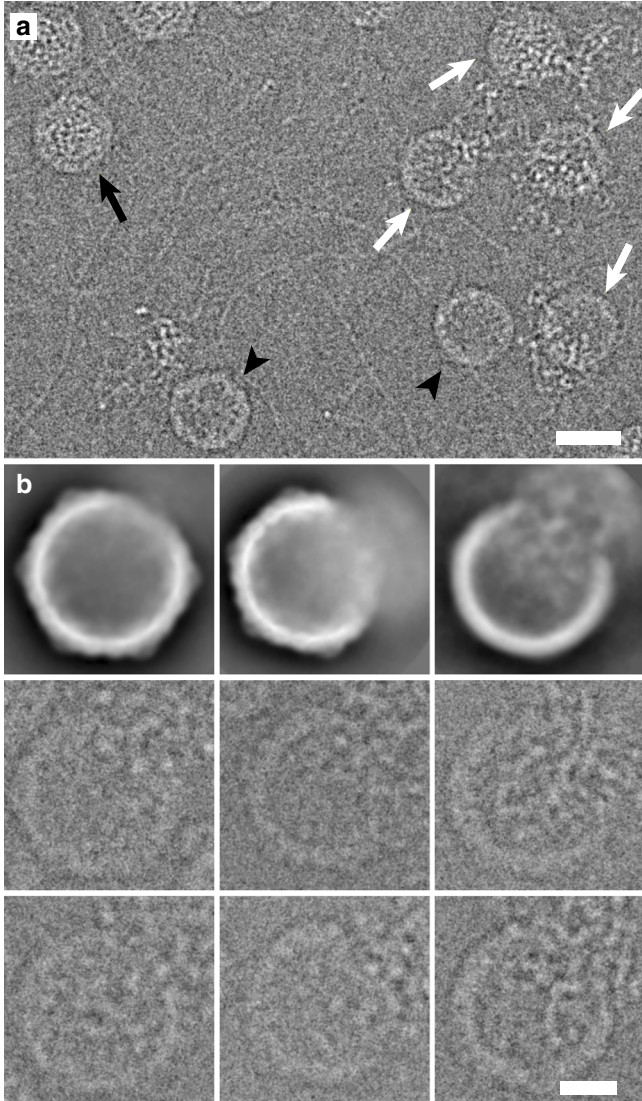

**Fig. 1** Particles of echovirus 18 in the process of genome release lack parts of their capsids. **a** Cryo-electron micrograph of echovirus 18 particles captured in the process of genome release after incubation at acidic pH. Black arrow indicates activated particle, white arrows particles in the process of genome release, and black arrowheads empty particles. Scale bar represents 25 nm. **b** Reference-free two-dimensional class averages showing echovirus 18 particles in the process of RNA release lacking parts of their capsids. Two representative electron micrographs of particles are shown for each class average. Scale bar represents 10 nm

**Re-structuring of echovirus 18 genome enables capsid opening**. Empty particles of echovirus 18 formed after genome release, induced by exposure to acidic pH for 2 h, were used to calculate their icosahedral reconstruction to a resolution of 3.2 Å (Supplementary Table 1, Supplementary Figs 4, 7). This is consistent with previous experiments, which show that empty capsids of enteroviruses are stable in vivo and in vitro[1–5,20]. Because the empty capsids of echovirus 18 are stable under the experimental conditions, the discharged pentamers can bind back to the capsid openings after the genome release[25]. It is likely that the re-assembly of the empty particles was favored by the high (0.5 mg/mL) concentration of echovirus 18 particles in samples that were prepared for cryo-EM observations. Nevertheless, the fate of the empty capsids after genome delivery is unimportant for the infection process in vivo.

The stability of echovirus 18 empty capsids under the conditions promoting genome release provides evidence that the force for the expulsion of pentamers from the activated particles is provided by the RNA genome. Comparing the cryo-EM images of native virions with activated particles of echovirus 18 (Fig. 5), as well as other enteroviruses[3,26,27], reveals that their genomic RNAs undergo conformational changes upon exposure to acidic pH. The conversion of echovirus 18 virions to activated particles occurred in <3 min after exposure to acidic pH at 37 °C, however some of the particles also released their RNA and aggregated (Supplementary Fig. 8). This rapid conversion to activated particles and genome release are consistent with previous experiments showing that human rhinovirus 2 delivers its genome into the cell cytoplasm within 2 min[28]. The electron density of the genomes is distributed uniformly in echovirus 18 virions at neutral pH, but transforms to a grainy appearance in activated particles at acidic pH (Fig. 5). During the shift from neutral to acidic pH, the side chains of histidines of capsid proteins (Supplementary Fig. 9), and probably also parts of the genomic RNA, become protonated, and thus acquire more positive charge. The reduction in the negative charge may disrupt interactions between the RNA and positively charged polyamines present within the virions[17,29,30]. The changes in the charge distribution and putative release of the polyamines from virions may result in the observed changes in the genome structure and increased pressure on the capsid from the inside. The inter-pentamer contacts of 11,000 Å² in native echovirus 18 virion are reduced to 5400 Å² in the activated particles. Correspondingly, the interaction between two pentamers in an activated particle of echovirus 18 is 25% weaker than that in a native virion (Supplementary Fig. 10). The weakening of the inter-pentamer contacts together with the changes in the genome organization, probably enable the opening of the activated particles for genome release.

**Molecular dynamics simulation of genome release**. The dynamics of the capsid opening and genome release were investigated using in silico simulations of a simplified model of the picornavirus capsid with its genome (Fig. 4). Using certain parameters in the model (for details, see Methods), we observed genome release after which a pentamer was separated from a capsid (Fig. 4, Supplementary Movie 1). In the simulations, the capsid first cracked open to allow the release of part of the genome. During this process, one or a few pentamers separated from the rest of the capsid. Subsequently, the two fragments of the capsid closed, resulting in an open capsid missing one or a few pentamers (Fig. 4). The pressure from the inside of the particle required to crack open the capsid is two-thirds of that required to expel a pentamer (for details, see Methods). Consequently, enterovirus capsids are more likely to rupture into two halves than to expel a single pentamer. After capsid rupture the genome pressure decreases, so that the two halves of the capsid may not separate completely and can quickly reassemble. The escaping genome can break off some pentamers, as observed in our cryo-EM experiments and simulations.

**Open particles were also observed for other enteroviruses**. Open particles are also present in the samples of echovirus 30 (Fig. 6). Furthermore, Harutyunyan et al. detected particles lacking pentamers in a sample of human rhinovirus 2 with crosslinked RNA genomes exposed to acidic pH[26]. These observations indicate that enteroviruses may release their genomes through openings formed by the removal of pentamers of capsid protein protomers from their capsids. Numerous capsid-binding antiviral compounds inhibit the genome release of

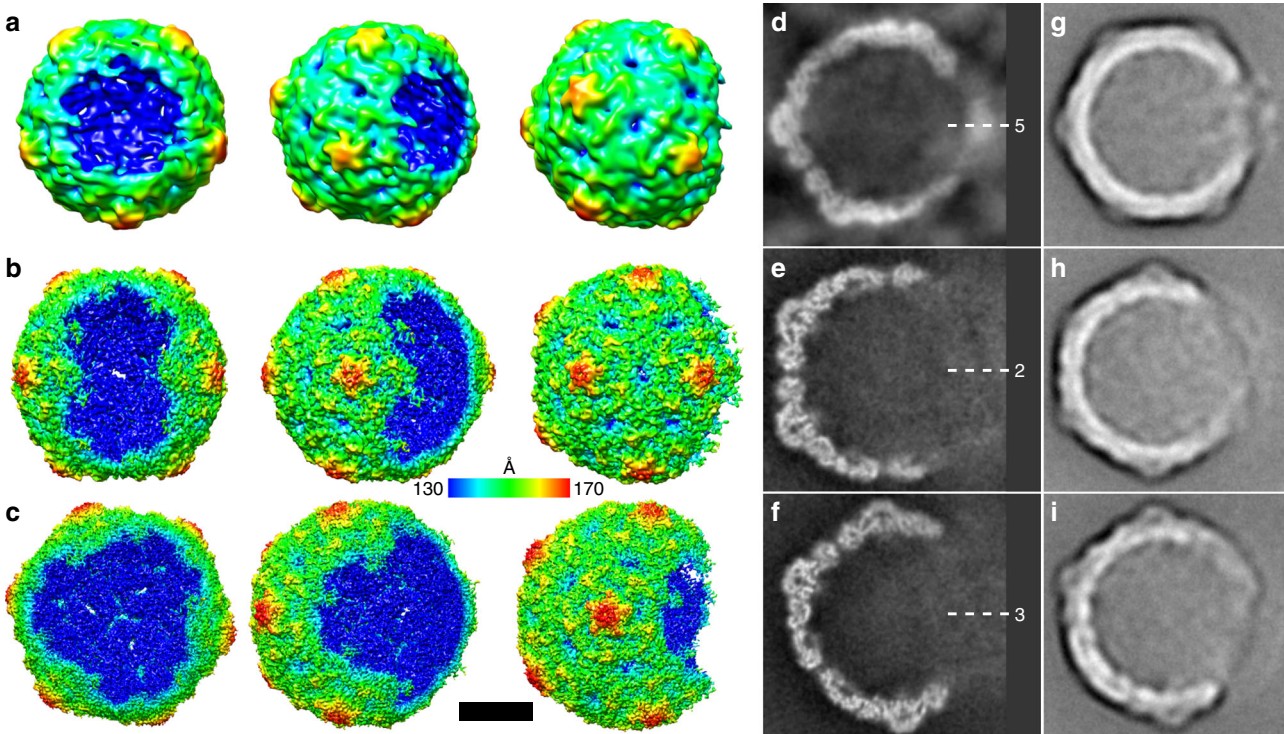

**Fig. 2** Open particles of echovirus 18 lacking one, two, or three adjacent pentamers. **a–c** Asymmetric three-dimensional reconstructions of open particles lacking one (**a**), two (b), or three (**c**) pentamers. The electron density maps are rainbow colored based on the distance of the electron density surface from the particle center. **d–f** Electron density distributions in central sections of asymmetric reconstruction of open particles missing one (**d**), two (**e**), or three (**f**) pentamers. The directions of five-fold, two-fold, and three-fold symmetry axes are indicated. Diffuse density in the areas of the missing pentamers probably belong to the average of the RNA molecules escaping from the particles. Example reference-free two-dimensional class averages of final three-dimensional refinement with C1 symmetry of open particles missing one (**g**), two (**h**), or three (**i**) pentamers. Scale bar represents 10 nm

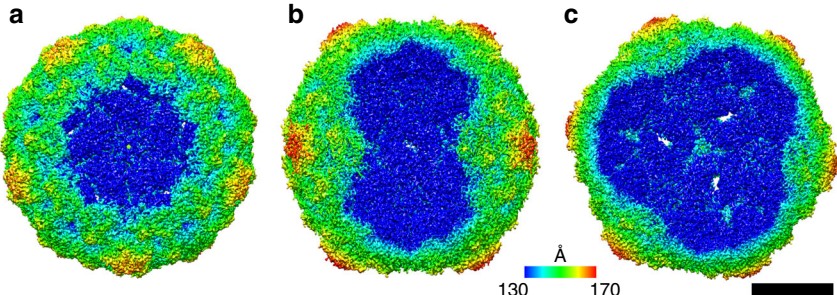

**Fig. 3** Symmetrized three-dimensional reconstructions of open particles. Echovirus 18 particles lacking one (**a**), two (**b**), or three (**c**) pentamers. Five-fold (**a**), two-fold (**b**), and three-fold (**c**) symmetries were employed during the reconstructions. The electron density maps are rainbow colored based on the distance of the electron density surface from the particle center. Scale bar represents 10 nm

enteroviruses[31–34] and an improved understanding of the process may facilitate drug development. In summary, our data show that enterovirus genome release requires several consecutive structural changes in a capsid (Fig. 7). Receptor binding or exposure to acidic pH induces the transformation of native virions into activated particles with reduced inter-pentamer interfaces[1,3,7–9,14,15]. The weakening of inter-pentamer contacts and changes in the genome structure enable opening of the capsids and release of the RNA genome.

## Methods

**Production and purification of echovirus 18.** Echovirus 18 (strain METCALF, ATCC-VR-852™) was propagated in immortalized African green monkey kidney (GMK, 84113001 Sigma) cells cultivated in Dulbecco's modified Eagle's medium

enriched with 10% fetal bovine serum. For virus preparation, 50 tissue culture dishes with a diameter of 150 mm of GMK cells grown to 100% confluence were infected with echovirus 18 with a multiplicity of infection of 0.01. The infection was allowed to proceed for 5–7 days, at which point more than 90% of the cells exhibited the cytophatic effect. The cell media were harvested and any remaining attached cells were removed from the dishes using cell scrapers. The cell suspension was centrifuged at 15,000 × $g$ in a Beckman Coulter Allegra 25R centrifuge, rotor A-10 at 10 °C for 30 min. The resulting pellet was re-suspended in 10 mL of phosphate-buffered saline (PBS) (10 mM Na$_2$HPO$_4$, 1.8 mM KH$_2$PO$_4$, 137 mM NaCl, and 2.7 mM KCl, pH 7.4). The solution was subjected to three rounds of freeze-thawing by transfer between −80 and 37 °C, and homogenized using a Dounce tissue grinder. Cell debris was separated from the supernatant by centrifugation at 3100 × $g$ in a Beckman Coulter Allegra 25R centrifuge, rotor A-10 at 10 °C for 30 min. The resulting supernatant was combined with media from the infected cells. Virus particles were precipitated by the addition of PEG-8000 and NaCl to final concentrations of 12.5% (w/v) and 0.6 M, respectively. The precipitation was allowed to proceed overnight at 10 °C and with mild shaking. The

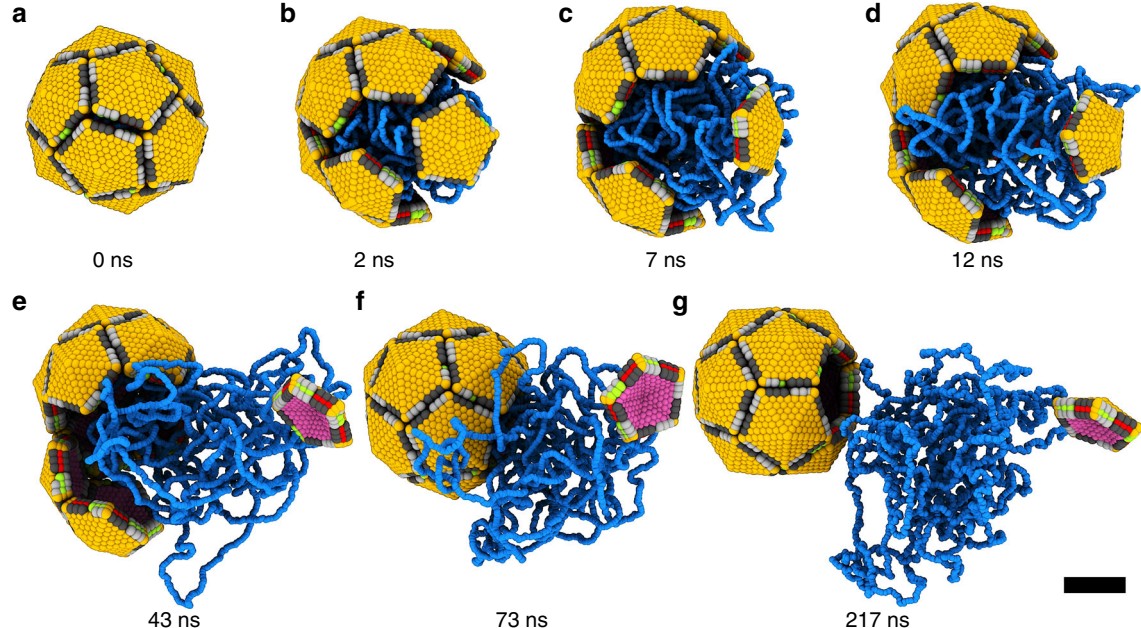

**Fig. 4** Molecular dynamics simulation of echovirus 18 genome release. Seven snapshots from the process were selected. **a** Compact capsid just before opening. **b** Initial cracking of the particle with one pentamer separated from the rest of the capsid. **c–e** Continued release of the genome. **f** Re-assembly of the capsid missing one pentamer. **g** Remainder of the genome diffuses from the open capsid. Genome is shown in blue, outer capsid surface in orange, inner capsid surface in purple, beads at pentamer edges shown in dark and light gray, green, and red represent attractive inter-pentamer interfaces. Scale bar represents 10 nm

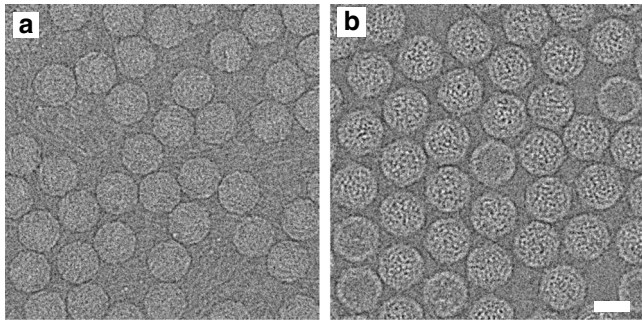

**Fig. 5** Exposure to acidic pH for 3 min at 4 °C induces conformational changes in echovirus 18 genome. **a** Electron micrograph of native echovirus 18 virions with uniformly distributed density of RNA genome. **b** Electron micrograph of echovirus 18 activated particles with grainy distribution of electron density belonging to RNA genome. Scale bar represents 25 nm

following day, the precipitate was centrifuged at 15,000 × g in a Beckman Coulter Allegra 25R centrifuge, rotor A-10 at 10 °C for 30 min. The pelleted white precipitate was re-suspended in 12 mL of PBS. MgCl$_2$ was added to a final concentration of 5 mM, and the sample was subjected to DNAse (10 μg/mL final concentration) and RNAse (10 μg/mL final concentration) treatment for 30 min at ambient temperature. Subsequently, trypsin was added to a final concentration of 0.5 μg/mL and the mixture was incubated at 37 °C for 10 min. EDTA at pH 9.5 was added to a final concentration of 15 mM and non-ionic detergent, NP-40$^{TM}$ (Sigma Aldrich Inc.), was added to a final concentration of 1%. The virus particles were pelleted through a 30% (w/v) sucrose cushion in re-suspension buffer (0.25 M HEPES, pH = 7.5, and 0.25 M NaCl) by centrifugation at 210,000 × g in an Optima X80 ultracentrifuge using a Beckman Coulter$^{TM}$ Ti50.2 rotor at 10 °C for 2 h. The pellet was re-suspended in 1.5 mL of PBS and loaded onto 60% (w/w) CsCl solution in PBS. The CsCl gradient was established by ultracentrifugation at 160,000 × g in an Optima X80 ultracentrifuge using a Beckman Coulter$^{TM}$ SW41Ti rotor at 10 °C for 18 h. The opaque bands containing the virus was extracted with a 20-gauge needle mounted on a 5 mL disposable syringe. The virus was transferred into 10 mM Tris, pH = 7.4, and 0.1 M NaCl by multiple rounds of buffer exchange using a centrifugal filter device with a 100-kDa molecular weight cutoff.

**Cryo-EM sample preparation and data acquisition**. For cryo-EM, 3.5 μL of echovirus 18 solution (2 mg/mL) were blotted and vitrified using a Vitrobot Mark IV on Quantifoil R2/1300 mesh holey carbon grids (vitrobot settings blot-force 2, blotting time 2 s). To observe echovirus 18 particles in the process of genome release, 5 μL of virus solution at a concentration of 2 mg/mL in 10 mM Tris, pH = 7.4, and 0.1 M NaCl was diluted in 15 μL of 50 mM Mes, pH 6.0. To obtain samples for single-particle analysis, the virus was incubated in acidic pH for 10 min at 4 °C. To measure the speed of formation of activated particles, the sample was incubated at 37 °C. Electron micrographs of virus particles were collected using an FEI Titan Krios transmission electron microscope operated at 300 kV. The sample in the column of the microscope was kept at −196 °C. Images were recorded with an FEI Falcon III direct electron detection camera under low-dose conditions (22.6 e$^-$/Å$^2$) with defocus values ranging from −1.0 to −3.0 μm at a nominal magnification of ×75,000, resulting in a pixel size of 1.061 Å/px. Each 1 s of exposure was recorded in movie mode and saved as 39 separate movie frames. The frames from each exposure were aligned to compensate for drift and beam-induced motion during image acquisition using the program motioncor2[35]. The resulting dose-weighted sum of aligned frames was used in the subsequent image processing steps, except for estimating contrast transfer function (CTF) parameters, which were determined from non-dose-weighted micrographs using the program gCTF[36].

**Single-particle reconstructions of echovirus 18**. Particles of echovirus 18 (512 × 512 pixels) were automatically selected from micrographs by Gautomatch. The images were processed using the package RELION 2.1[37]. The dataset of autopicked particles of echovirus 18 was subjected to 2D classification. Classes containing full and empty particles were selected for processing in parallel. Classes containing full particles were 3D-classified to obtain sets of native particles and activated particles. Echovirus 7 [PDB ID: 2X5I] served as the initial model for these classifications[38]. Similarly, classes containing empty particles were subjected to 3D classification to obtain a homogeneous set of empty particles. Finally, refinement of the selected particles was performed for native, activated particles and empty particles, using the RELION 3dautorefine procedure[37]. Icosahedral symmetry was imposed on the volumes during the refinement process.

**3D reconstructions of open particles**. Schemes of the image processing, classification and reconstruction of echovirus 18 and echovirus 30 are shown in Supplementary Fig. 2. Particles of echovirus 18 (512 × 512 pixels) were automatically selected by Gautomatch. Particles of echovirus 30 in the process of genome release (512 × 512 pixels) were manually boxed using the program e2boxer.py from EMAN2[39]. The images were processed using RELION 2.1[37]. The dataset of autopicked echovirus 18 articles was subjected to 2D classification. Classes containing particles releasing their genomes were selected for further 3D classification.

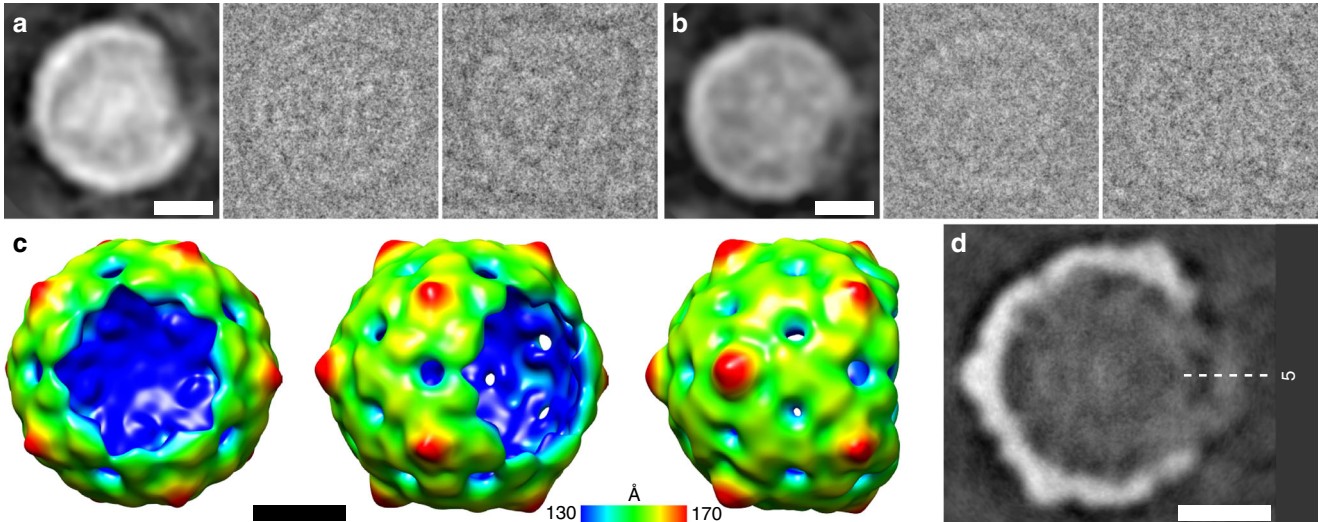

**Fig. 6** Open particle of echovirus 30. **a, b** Reference-free two-dimensional class averages showing echovirus 30 particles lacking parts of their capsids. For each class, average images of two representative particles are shown. **c** Three-dimensional reconstruction of open particle of echovirus 30 lacking one pentamer with imposed five-fold symmetry. The electron density map is rainbow colored based on the distance from the particle center. **d** Electron density distribution in central section of reconstruction of open particle. The diffuse density in the pore formed by the missing pentamer probably belongs to the average of RNA molecules escaping from the particles. Scale bars represent 10 nm

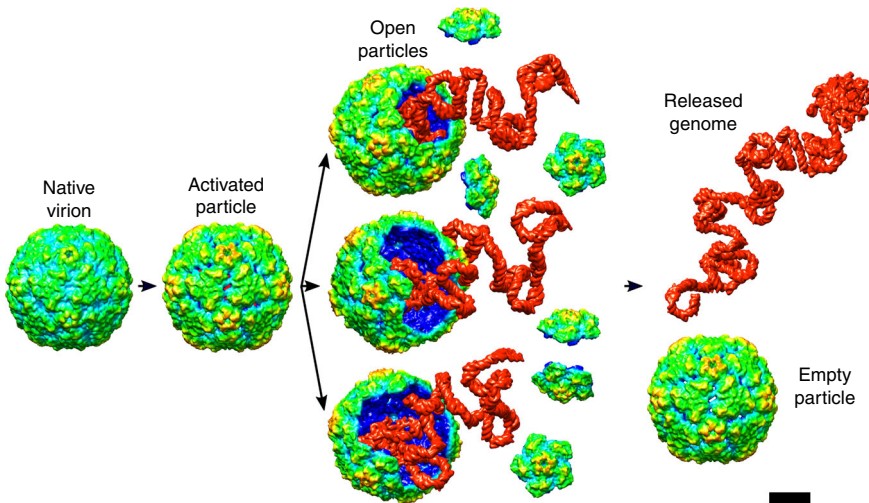

**Fig. 7** Scheme of enterovirus genome release. Binding to receptors or exposure to acidic pH in endosomes induces conformational transition of virions to activated particles. The structural changes within the capsid and virus RNA enable the expulsion of pentamers from the capsid, resulting in the formation of open particles. The RNA genomes are released from the open particles. After the genome release, the pentamers may re-associate with the open capsids. Scale bar represents 10 nm

The structure of the echovirus 18 or echovirus 30 "A" particle, low-pass filtered to a resolution of 50 Å, was used to initiate asymmetrical 3D classification of echovirus 18 and echovirus 30, respectively. Classes with particles exhibiting defects corresponding to missing one, two, or three pentamers were selected for further 3D classification. In these classification steps and in the final refinement step, the previous 3D class reconstructions served as the initial models. These reconstructions were rotated so that the symmetry axes of the open particles were aligned with the z-axis. The resulting 3D classes with homogenous particles missing one, two, or three pentamers were independently refined using the RELION 3dautorefine procedure as asymmetric reconstructions or by imposing the appropriate symmetries.

**Post-processing of the refined maps**. The volumes resulting from the 3D refinement were threshold masked, detector modulation transfer function corrected and B-factor sharpened using the RELION postprocess procedure. To avoid over-masking, the masked maps were visually inspected to exclude the possibility of the clipping of electron densities belonging to the virus capsid. Additionally, the occurrence of over-masking was monitored by inspecting the shapes of Fourier shell correlation (FSC) curves. Furthermore, the shapes of the FSC curves of phase-randomized half-datasets with the applied mask were checked. The resulting resolutions of the reconstructions were estimated as the values at which the FSC curves fell to 0.143.

**Structure determination of open particles of echovirus 18**. The initial model, the activated particle of echovirus 18, was rigid-body fitted into the B-factor-sharpened cryo-EM map of echovirus 18 open particles and subjected to manual rebuilding using Coot[40], and coordinate and B-factor refinement using Phenix[41].

**Charge calculation - Monte Carlo simulations**. We performed Metropolis Monte Carlo (MC) simulations using the Faunus framework[42]. The spherical cell with a radius of 45 nm contained one copy of the capsid described with an implicit-solvent coarse-grained model, where every residue was treated as a spherical bead

(located at the center of mass of the residue) with a radius derived from the amino-acid molecular weight and the common density of 0.9 g/mL. The N- and C termini of both proteins were represented as separate residues. The solvent was treated as a dielectric continuum using the Debye–Hückel approximation with a relative permittivity of 78.7 for the interaction of charged residues[43,44]. The capsid was placed in the middle of the simulation sphere with all degrees of motion frozen. Each amino acid was allowed to change its protonation state by titration move, where protons are allowed to move between the bead and solution. The energy associated with the exchange is determined by the change in local electrostatic energy $\pm$ (pH − p$K_0$)ln10, where p$K_0$ is the dissociation constant of the isolated amino acid, and pH is that of the system[45]. The plus and minus in the equation is associated with protonation and de-protonation, respectively. Titratable residues with their pKa values are: C terminus (2.6), Asp (4.0), Glu (4.4), His (6.3), N-terminus (7.5), Tyr (9.6), Lys (10.4), Cys (10.8), and Arg (12.0). The total number of moves, in which there are attempts to protonate/deprotonate residues, was at least 1000 per each residue in all simulations. The temperature of our NVT ensemble was set to 298 K. We performed calculations of capsids of both native virions and activated particles with structures determined from cryo-EM. The average charges of amino acids were determined for pH = 7.4 and 6.0, and monovalent salt solutions of concentrations 150 and 40 mM.

**Comparison of forces required to open a capsid**. The reason why the capsids are more likely to initially crack open rather than directly expel pentamers can be shown by a comparison of the forces holding the capsid together to the force exerted by the genome, which acts to rupture the capsid from the inside. Assuming that the inside pressure generated by the genome is homogeneous, the force acting on each pentamer is proportional to its area of 1/12 of the sphere surface: $f_{pentamer} = \pi r^2 \, p/3$, where $r$ is the capsid radius and $p$ is the excess pressure from inside to outside. Each pentamer in the capsid interacts with five others, generating a force of 5 F that holds the pentamer in the capsid. Therefore, the pressure of the genome required to break a pentamer away is $p = 15 \, F/(\pi r^2)$. However, the pressure to separate two halves of a capsid is only $p = 10 \, F/(\pi r^2)$, because two half-capsids interact with each other through 10 inter-pentamer interfaces, resulting in a holding force of 10 F and the pressure force exerted by the genome is $f_{half} = \pi r^2 \, p$ due to the half-capsid projection in the direction of the force.

**Molecular dynamics simulations**. All-atom molecular dynamics simulations were performed using GROMACS version 2016.4[46,47]. The initial structure from cryo-EM was minimized and equilibrated for 10 ns using the all-atom Amber99SB-ILDN force field[48]. During the all-atom equilibration, the system was kept in an isothermal-isobaric ensemble using position restraints on backbone atoms. The spring constant of the position restraints was 1000 kJ/mol/nm². Temperature was held at 300 K with a velocity-rescaling thermostat[49]. The time constant for temperature coupling was set to 0.1 ps. We employed an isotropic Berendsen barostat[50] set to 1 bar using a 1 ps coupling constant. Lennard-Jones forces were calculated with a cutoff radius of 1.1 nm. The same cutoff was applied for real-space electrostatic interactions, while the long-range contribution was evaluated with particle mesh Ewald[51]. All bonds were constrained using the LINCS algorithm[52], apart from TIP3P water, for which analytical SETTLE was used[53]. A 2-fs time step was used for the production run. The system consisted of four echovirus 18 protomers in aqueous solution (of roughly 200,000 water molecules) with added NaCl ions at a concentration of 150 mM. We simulated capsids of both native virions and activated particles in a rectangular box of 16 × 16 × 21 nm and 17 × 17 × 25 nm, respectively. Periodic boundary conditions were applied.

For the binding free energy calculations between two pentamers, we used a computationally efficient, coarse-grained MARTINI 2.2 force field[54–56]. The resulting structure from all-atom equilibration was converted into a MARTINI model using the *martinize.py* script. As a consequence of coarse-graining, the MARTINI model does not explicitly describe backbone hydrogen bonds. Thus, the secondary structure has to be imposed on the peptides and maintained throughout the simulation. The assignment of secondary structure for both native virion and activated particle echovirus 18 was done with the program DSSP[57]. To help preserve the higher-order structure, an elastic network was added to the standard MARTINI topology. Harmonic bonds were generated between backbone beads by the *martinize.py* script using the option -*elastic*. The elastic bond force constant was set to 500 kJ/mol/nm² (−*ef 500*), and the lower and upper elastic bond cutoff radius to 0.5 and 0.9 nm (−*el 0.5* and −*eu 0.9*), respectively, and the elastic bond decay factor and decay power both to 0 (−*ea 0* and −*ep 0*). Furthermore, elastic bonds were deleted for residues exhibiting a high degree of flexibility in the electron density map (Supplementary Table 3). Additionally, the mapping of histidines with an average charge over 0.4 e, determined from MC simulations, were changed from the uncharged to charged form. The simulation time step was set to 30 fs. A velocity-rescaling thermostat with a coupling constant of 1.0 ps was employed to maintain the temperature at 310 K[49]. Protein and solvent beads were coupled to separate baths to ensure the correct temperature distribution. The pressure was kept at 1 bar with a Parrinello-Rahman barostat with an isotropic coupling scheme with a coupling constant of 12 ps[58]. All non-bonded interactions were cutoff at 1.1 nm and the van der Waals potential was shifted to zero. The relative dielectric constant was set to 15. Periodic boundary conditions were employed, yielding a rectangular box of dimensions 17.7 × 17.7 × 28.5 nm for the activated particle and

17.7 × 17.7 × 31.4 nm for the native virion. The System consisted of 4 protomers of capsid proteins of echovirus 18 in water with added NaCl ions at a concentration of 150 mM.

The umbrella sampling method was employed to determine the free energy of binding between two pentamers. We defined the reaction coordinate as the $z$-distance between the centers of mass of protomers 1–2 and 3–4 (Supplementary Fig. 11). We restrained the position of protomers 1–2 through the use of harmonic potentials on backbone beads, excluding the flexible residues from Supplementary Table 3. Cylindrical flat-bottomed position restraints were applied to the backbone beads of protomers 3–4, excluding the residues from Supplementary Table 3, to keep the protomers from tilting and moving in the $XY$ plane. The cylinders were parallel to the $z$-axis. The force constant was set to 1000 kJ/mol/nm² and the radius of all cylinders was 0.3 nm. The reference configuration for the cylindrical flat-bottomed position restraint was selected from a 1000-ns equilibration run. For the last 500 ns of the equilibration run, the structure of protomers 3–4 was averaged. The reference configuration was selected from the trajectory based on the lowest root mean squared deviation toward the averaged structure. For the native virion, 74 umbrella windows were simulated for 2000 ns each, which was necessary to get a convergence (Supplementary Fig. 12). The spring constant of the umbrella harmonic potential was set to 50,000 kJ/mol/nm² for the first 40 windows, with a spacing of 0.02 nm. The next 34 windows were spaced by 0.025 nm and the spring constant was set to 10,000 kJ/mol/nm². For the activated particle of enterovirus 18, 149 umbrella windows were simulated for 1000 ns each. The first 34 windows were spaced by 0.02 nm and a harmonic spring of 50,000 kJ/mol/nm² was applied. The next 115 windows had the spring constant set to 10,000 kJ/mol/nm² with a spacing of 0.025 nm. To analyze the probability distributions of states from each window, iterative WHAM was used, implemented in the GROMACS tool *gmx wham*[59,60].

**Phenomenological model**. We developed a phenomenological coarse-grained model of a *Picornavirus* family based on human echovirus 18 (Supplementary Fig. 13). The capsid was a regular dodecahedron comprised of 12 pentagonal subunits. Each subunit assumes the role of a stable pentameric intermediate. The pentamers were made of beads (pseudoatoms) organized in three layers.

The outer circumscribed sphere of the capsid had a radius of 16 nm and the inner was 14 nm. Each pentamer was composed of 317 beads connected with 1311 harmonic bonds keeping the structure. The spring constant of the harmonic potential was 250 kJ/mol. Beads within the capsids were only interacting via harmonic bonds. There were six types of beads (Supplementary Fig. 14), all of which were interacting with Weeks-Chandler-Anderson repulsive potential with an epsilon set to 1.0 [https://doi.org/10.1063/1.1674820]. In addition, beads at the pentamer edges had an attractive interaction range of 0.3 to 2.0 nm based on the free energy calculation with the Martini model. The interaction decreased to zero with a cos² dependence. The attraction strength was weak, 0.5 kJ/mol, for the inner and outer layer, while the middle layer had stronger attraction strength that varied from 2 to 20 kJ/mol. The attraction only acts between the types, which are in contact in the assembled capsid structure. These interactions represent specific contacts between the protomers in the capsid (Supplementary Fig. 11).

To investigate RNA genome release from the capsid, we modeled the RNA as a chain of repulsive beads. A single bead represented two nucleotides with a radius of 0.6 nm connected by a 1.1-nm-long harmonic bond, which is about twice the distance between phosphates of adjacent nucleotides. All beads were interacting with a shifted truncated Lennard-Jones potential, i.e. Weeks-Chandler-Anderson potential with an epsilon set to 1.0[61].

Simulations were performed in LAMMPS[62], with the use of a Langevin thermostat[63–65]. Center of mass motion of the entire system caused by the thermostat was eliminated using the option "zero yes". Additionally, the "gjf yes" option was turned on, applying Gronbech-Jensen/Farago time-discretization for the Langevin model to enable longer time steps, while still producing the correct Boltzmann distribution of atom positions[65]. The viscous damping term was set to 10,000 time steps. The reduced temperature in our simulations was $T^* = 1 \, k_B T$. The box size 150 × 150 × 150 nm was constant and the same for all simulations.

The simulation protocol was as follows. First the capsid was generated from pentamers, and then the chain representing the genome was generated within the capsid using a random walk. The equilibration started with the chain equilibration alone, which was simulated with Langevin dynamics for 10⁸ steps, while the capsid shell was motionless. The second step was the equilibration of both the capsid and genome. The attraction between pair B-C begun at 35 kJ/mol and was gradually decreased by a rate of 0.25 kJ/mol every 200,000 time steps. The production genome release simulations were repeated 10 times, with different conditions, for each set of parameters (attraction strength, attraction range, flexibility of the capsid, and the size of the genome beads) each of which were concluded at 10⁹ time steps or sooner if the release had occurred.

To estimate the timescale of the simulations, we performed 50 independent simulations of the full virus capsid starting from different conditions. Each simulation was performed for 10⁸ steps. The parameters for the capsid were chosen to prevent the genome release within the duration of the simulation. We analyzed the averaged square displacement of the full capsid center of mass. Using Fick's second law, we calculated the diffusion coefficient of the capsid to be $1.37 \times 10^{-4} \pm 2.0 \times 10^{-8}$ Å²/step (Supplementary Fig. 15). We also estimated the diffusion coefficient using the Stokes-Einstein relation, with a capsid radius of 16.5 nm, the

diffusion coefficient is 15 μm²/s. Comparing the estimated coefficient with the simulation one, the simulation time step corresponded to 92 fs (there were 10,901 ± 1 steps in a ns).

**Timescale of genome release**. To estimate the timescale of genome release from the virus capsid, we analyzed the average times of this process in our simulations. The genome was released from the capsid in the order of $10^6$ simulation steps corresponding to 100 ns. Another approach is to estimate the time required for the genome to diffuse freely from an open capsid. Using Einstein-Stokes relation, we determined that the genome diffuses 50 nm in microsecond timescale, which makes the observation of the genome release experimentally challenging.

**Reporting summary**. Further information on experimental design is available in the Nature Research Reporting Summary linked to this article.

## Data availability

Cryo-EM electron density maps have been deposited in the Electron Microscopy Data Bank, https://www.ebi.ac.uk/pdbe/emdb/ (accession numbers EMD-0181, EMD-0182, EMD-0183, EMD-0184, EMD-0185, EMD-0186, EMD-0187, EMD-0188, EMD-0189, and EMD-0217), and the fitted coordinates have been deposited in the Protein Data Bank, www.pdb.org (PDB ID codes 6HBG, 6HBH, 6HBJ, 6HBK, 6HBL, and 6HHT). The authors declare that all other data supporting the findings of this study are available within the article and its Supplementary Information files, or are available from the authors upon request.

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

## Acknowledgements

We wish to thank the Central European Institute of Technology Core Facilities of Cryo-Electron Microscopy and Tomography and of Proteomics, supported by the Czech Infrastructure for Integrative Structural Biology (LM2015043 funded by the Ministry of Education, Youth and Sports of the Czech Republic) for their assistance in obtaining the scientific data presented in this paper. This research was carried out under the project CEITEC 2020 (LQ1601), with financial support from the Ministry of Education, Youth and Sports of the Czech Republic under National Sustainability Program II. This work was supported by The Ministry of Education, Youth and Sports from the Large Infrastructures for Research, Experimental Development and Innovations project "IT4Innovations National Super-computing Center – LM2015070". Computational resources were provided by CESNET LM2015042, and the CERIT Scientific Cloud LM2015085 provided under the program Projects of Large Research, Development, and Innovations Infrastructures of The Ministry of Education, Youth and Sports. The Titan Xp used for this research was donated by the NVIDIA Corporation. The research leading to these results received funding from the European Research Council under the European Union's Seventh Framework Program Grant (FP7/2007-2013)/ERC Grant Agreement 335855 (to PP), Czech Science Foundation Grant Agreement GX19-25982X (to P.P.), and EMBO Grant Agreement IG 3041 (to P.P.).

## Author contributions

P.P. and R.V. designed research; D.B., T.F., D.H., Y.L., L.S., L.M. and J.M. performed research; D.B., T.F., D.H., L.S., R.V. and P.P. analyzed data; and D.B., T.F., L.S., R.V. and P.P. wrote the paper.

## Additional information

**Competing interests:** The authors declare no competing interests.

