## [Peer Review File · Nature Communications]

Reviewers' Comments:

Reviewer #1:

Remarks to the Author:

In this brief manuscript Buchta, et al, describe in molecular detail Coxsackievirus particles in the process of shedding their RNA content as a consequence of exposure to acidic conditions. They show images of 'activated' particles in which the diffuse density in the centre of the virions and attributable to RNA has taken on a grainy appearance following exposure to pH6.0, so revealing conformational changes in the organisation of the genome. The genome 'shedding' particles have lost between one and three pentameric subunits to produce a substantial opening through which density that the authors ascribe to exiting RNA can be seen. These structures are compelling and if relevant to the cell entry process they address the problem of how large secondary and tertiary structure elements in the viral genome can exit the particle since the holes produced through shedding pentameric subunits are substantially larger than the apertures that have previously been reported as potential RNA exit sites. However, their interpretation of the RNA release process involving the reversible loss of pentameric subunits contrasts with the (potentially) conflicting model in which a portal is assembled at the endosomal membrane from the internal protein VP4 and components of the VP1 protein which allows protected transfer of the genome into the host cell. The mechanistic details of the picornavirus entry process are still unclear and it is important for the research community to consider all potential possibilities such as those described in this manuscript.

The results described here are important and interesting but there is a lot to be done before the relevance of their observations to the infection process can be properly evaluated. Some of the many questions to be addressed in future studies are; how do their observations relate to the well known entry intermediate 'A' particles which have shed the internal VP4 protein but still retain the RNA?; what is the role, if any, of the virus receptor in the process?; how does this mode of genome release allow genome transfer across membranes?; how can the terminus specific release of the genome (for rhinovirus) be explained in this model?; etc.

I have few criticisms of the study as presented but the following comments should be addressed: -

1) There is very little information on the dynamics of the genome release mechanism they describe. 'Activated' particles in which the RNA has acquired a granular appearance are described but I could not see a statement of the time of exposure to acid pH necessary for this conversion. On the other hand the micrographs showing particles with missing subunits and in the process of shedding RNA were exposed to acid for 2 hours. As the process of cell entry is rapid this seems a rather long time of exposure.

2) The end products of genome release for enteroviruses are 80S empty particles. It is therefore assumed that the release of capsid subunits is a reversible process. If so I would expect the reassembly process to be time and concentration dependent. Have the authors considered investigating these questions?

Reviewer #2:

Remarks to the Author:

The manuscript by Buchta et al. (NCOMMS-18-26966-T) shows the cryoEM structures of several forms of the Echovirus 18 virus particle, including the virion, the "activated particle" and the empty capsid at about 3Å resolution that are apparently similar to the equivalent particles (virions, A particles, B particles) described previously for other enteroviruses (e.g. PV, HRV14, HRV2 or EV71) but the manuscript in its present form does not contain any kind of comparison with the other related structures of uncoating intermediates existing in the literature.

What is really new in this manuscript is the description of novel uncoating intermediates in the process of genome release, lacking one, two or three capsid pentamers; the so called "open particles". These

new cryo-EM structures that are solved at modest resolution 6-9Å, have not been described so far in other enteroviruses. Authors also claim that this is a conserved mechanism of enterovirus genome release, but my main question is what are the experimental evidences that support this conclusion? Cryo-electron tomography studies by Bostina et al., showed a complete spherical capsid during genome release in PV (J Virol 2011; reference 8 in the manuscript). Also a number of studies on HRV2, CVA7 or EV71 uncoating intermediates show in all cases complete capsids (references 2-7).

Authors should at least discuss the differences observed between echovirus 8 and the rest of known enterovirus structures of uncoating intermediates trying to understand why echovirus 8 has a different behavior. It would be useful to have, a careful comparative analysis of the inter-pentameric interactions between echovirus 8 and other enteroviruses of known structures.

This reviewer has also a number of concerns related to different technical aspects

I am a bit surprised that in the micrograph shown in figure 1, containing: active particles, empty particles and also particles releasing the RNA, all the particles releasing RNA are on the same side of the micrograph. Taking into account that to study the particles releasing RNA, samples should be diluted 4 times, I am wondering if the gradient of vitreous ice in the grid hole can exert any force on the particles frozen in the central region, making them to collapse. To exclude this, it would be nice to know if the particles releasing RNA are localised preferentially in the center of the grid holes.

Figure 2 shows the reconstructions calculated for each of the 3 proposed models. Authors also show central sections of each reconstruction. I wonder if the presence of RNA could be masking what is really happening in the capsid and what they are seeing really is a capsid with the RNA region removed. To clarify this point, it would be of great help to have the reference-free two dimensional classes of the final 3D refinement with C1 symmetry for each of the three RNA releasing reconstructions.

In the 3D classifications, as well as in the 3D refinement, authors are using the models (lacking one, two or three pentamers), obtained in the prior 3D classification as initial models for the next steps. This fact could be introducing a bias in the reconstructions, specially if these reconstructions have not enough representative views. To discard this possible bias, It would be good to see the angle distribution of the particles.

Coordinates and cryo-EM maps should be deposited

Response to reviewers' comments

Reviewers' comments are in blue italics, our responses in black bold font.

Reviewer #1 (Remarks to the Author):

In this brief manuscript Buchta, et al, describe in molecular detail Coxsackievirus particles in the process of shedding their RNA content as a consequence of exposure to acidic conditions. They show images of 'activated' particles in which the diffuse density in the centre of the virions and attributable to RNA has taken on a grainy appearance following exposure to pH6.0, so revealing conformational changes in the organisation of the genome. The genome 'shedding' particles have lost between one and three pentameric subunits to produce a substantial opening through which density that the authors ascribe to exiting RNA can be seen. These structures are compelling and if relevant to the cell entry process they address the problem of how large secondary and tertiary structure elements in the viral genome can exit the particle since to holes produced through shedding pentameric subunits are substantially larger than the apertures that have previously been reported as potential RNA exit sites. However, their interpretation of the RNA release process involving the reversible loss of pentameric subunits contrasts with the (potentially) conflicting model in which a portal is assembled at the endosomal membrane from the internal protein VP4 and components of the VP1 protein which allows protected transfer of the genome into the host cell. The mechanistic details of the picornavirus entry process are still unclear and it is important for the research community to consider all potential possibilities such as those described in this manuscript.

The results described here are important and interesting but there is a lot to be done before the relevance of their observations to the infection process can be properly evaluated. Some of the many questions to be addressed in future studies are; how do their observations relate to the well known entry intermediate 'A' particles which have shed the internal VP4 protein but still retain the RNA?; what is the role, if any, of the virus receptor in the process?; how does this mode of genome release allow genome transfer across membranes?; how can the terminus specific release of the genome (for rhinovirus) be explained in this model?; etc.

I have few criticisms of the study as presented but the following comments should be addressed:

1) There is very little information on the dynamics of the genome release mechanism they describe. 'Activated' particles in which the RNA has acquired a granular appearance are described but I could not see a statement of the time of exposure to acid pH necessary for this conversion. On the other hand the micrographs showing particles with missing subunits and in the process of shedding RNA were exposed to acid for 2 hours. As the process of cell entry is rapid this seems a rather long time of exposure.

A: The 2 hours incubation at acidic pH was used to prepare empty particles of enterovirus 18 not the open particles (lines 131-133):

"Empty particles of echovirus 18 formed after genome release, induced by exposure to acidic pH for two hours, were used to calculate their icosahedral reconstruction to a resolution of 3.2 Å (STable 1, S Figs. 4,7)."

The formation of open particles was induced by exposure of echovirus 18 virions to acidic pH for 10 minutes. These experiments were performed at 4°C. This information has now been emphasized in the manuscript (lines 72-74):

“Specifically, we performed cryo-EM of echovirus 18 virions exposed to pH 6.0 for 10 minutes, mimicking the acidic environment that the virus encounters in endosomes (Fig. 1a, SFig. 1).”

The same information is included in Supplementary Materials and Methods section (lines 197-199):

“To obtain samples for single-particle analysis, the virus was incubated in acidic pH for 10 minutes at 4°C.”

To address reviewers comment we have now characterized the dynamics of conversion of E18 virions to activated particles at 37°C. We recorded cryo-EM data on E18 samples exposed to Mes pH 6.0 at 37°C for 3, 12 and 21 minutes. E18 virions converted to activated particles and started to release their genomes already within the 3 min exposure to the acidic pH. This is consistent with previous analyses showing that about 2 minutes are required for release of human rhinovirus 2 genomes *in vivo*. However, most of the particles incubated in Mes pH 6.0 at 37°C aggregated and would not be suitable for cryo-EM analyses. The results have now been included in the manuscript in SFig. 8 and described in the text (lines 147-152):

“The conversion of echovirus 18 virions to activated particles occurred in less than three minutes after exposure to acidic pH at 37°C, however some of the particles also released their RNA and aggregated (SFig. 8). This rapid conversion to activated particles and genome release are consistent with previous experiments showing that human rhinovirus 2 delivers its genome into the cell cytoplasm within 2 minutes²⁸.”

Fig. 5. showing comparison of native virions and activated particles of enterovirus 18 lacked the information about the time of incubation at acidic pH. This has now been corrected (lines 270-271):

“Exposure to acidic pH for 3 minutes at 4°C induces conformational changes in echovirus 18 genome.”

2) The end products of genome release for enteroviruses are 80S empty particles. It is therefore assumed that the release of capsid subunits is a reversible process. If so I would expect the reassembly process to be time and concentration dependent. Have the authors considered investigating these questions?

A: In agreement with reviewer #1 we expect that the reassembly of E18 capsids after genome release is concentration dependent. The speed of the reassembly is probably determined by the diffusion rate of the components. Obtaining sufficient cryo-EM data on samples with various particle concentrations at various time points of the reaction to validate this hypothesis is not feasible due to the limited time available on electron microscopes. Furthermore, the re-assembly of the empty capsids is most likely irrelevant for the infection process *in vivo*.

Discussion of the putative re-assembly of empty capsids has now been included in the manuscript (lines 137-141):

“It is likely that the re-assembly of the empty particles was favored by the high (0.5 mg/ml) concentration of echovirus 18 particles in samples that were prepared for cryo-EM observations. Nevertheless, the fate of the empty capsids after genome delivery is unimportant for the infection process *in vivo*.”

Reviewer #2 (Remarks to the Author):

The manuscript by Buchta et al. (NCOMMS-18-26966-T) shows the cryoEM structures of several forms of the Echovirus 18 virus particle, including the virion, the “activated particle” and the empty capsid at about 3Å resolution that are apparently similar to the equivalent particles (virions, A particles, B particles) described previously for other enteroviruses (e.g. PV, HRV14, HRV2 or EV71) but the manuscript in its present form does not contain any kind of comparison with the other related structures of uncoating intermediates existing in the literature.

A: Description of previous structural studies of enterovirus genome uncoating has now been included in the introduction and through the manuscript (lines 42-68):

“The interactions of enteroviruses with receptors or exposure to acidic pH in endosomes induce conformational changes in virions into an activated state characterized by increased particle diameter, reduced contact areas between pentamers of capsid protein protomers, release of VP4 subunits from particles, and externalization of the N-termini of VP1 subunits¹⁻⁵. The activated particles of numerous enteroviruses were shown to contain openings along twofold ($5 \times 10 \text{ \AA}$) or fivefold (diameters of up to 8 \AA) axes of icosahedral symmetry of their capsids²⁻⁶. It has been speculated that these pores serve for the release of enterovirus genomes^{3,6-10}. However, capsids of viruses from the families *Reoviridae* and *Totiviridae* that release single-stranded RNAs as part of their replication cycles contain circular pores larger than 15 \AA in diameter^{11,12}. The size of the pores in enterovirus capsids is not sufficient for the passage of single-stranded RNA^{2-6,13-15}. Furthermore, enterovirus genomes contain sequences that form double-stranded RNA segments which fold into three-dimensional structures, such as the internal ribosomal entry site required to initiate translation of viral RNA¹⁶. If these double-stranded RNA segments were present inside enterovirus particles, then the genome release would require either the opening of pores larger than 40 \AA in diameter, or a mechanism to unwind the double-stranded RNA. However, there is no evidence of an association between enzymes with RNA helicase activity and enterovirus virions^{1,17}. The structures of enterovirus particles before and after genome release have been characterized at high resolution^{3,6-10}. Most of these cryo-EM and X-ray crystallography studies imposed icosahedral symmetry during the structure determination process and were not aimed at identifying the unique site of genome exit^{3-7,9}. Asymmetric single-particle reconstruction and sub-tomogram averaging studies, at a resolution of 50 \AA , were used to indicate that RNA exits poliovirus particles close to a twofold axis⁸. The end stage of the enterovirus genome release are the empty capsids, the structures of which were determined for several enteroviruses^{4,5}.”

What is really new in this manuscript is the description of novel uncoating intermediates in the process of genome release, lacking one, two or three capsid pentamers; the so called “open particles”. These new cryo-EM structures that are solved at modest resolution 6-9Å, have not been described so far in other enteroviruses.

A: We would like to point out that asymmetric structures of the open particles were determined to the resolutions of 6-9 Å. The corresponding structures with imposed symmetry (5, 2, and 3 fold, respectively) were determined to the resolutions of 3.8, 4.1, and 3.7 Å. Display of the high-resolution reconstructions has now been included in the manuscript as Fig. 3.

Authors also claim that this is a conserved mechanism of enterovirus genome release, but my main question is what are the experimental evidences that support this conclusion? Cryo-electron tomography studies by Bostina et al., showed a complete spherical capsid during

genome release in PV (J Virol 2011; reference 8 in the manuscript). Also a number of studies on HRV2, CVA7 or EV71 uncoating intermediates show in all cases complete capsids (references 2-7). Authors should at least discuss the differences observed between echovirus 18 and the rest of known enterovirus structures of uncoating intermediates trying to understand why echovirus 18 has a different behavior.

A: We moderated our statements through the manuscript to avoid the statement “conserved mechanism”. Nevertheless, our claim that formation of open particles is a general mechanism of the genome release among enteroviruses is based on the observation of open particles for not only echovirus 18, but also echovirus 30 and human rhinovirus 2. We have now moved figure showing the open particle of echovirus 30 into the main text (Fig. 6).

We have now included discussion of the previous results on enterovirus genome release in the manuscript (lines 94-98):

“In contrast to the formation of the open particles of echovirus 18, complete capsids were observed in the cryo-EM study of the genome release of poliovirus⁸. However, the poliovirus uncoating was induced by the exposure of particles to 56°C, which may have affected the secondary structure of the genome and the mechanism of its release.”

It would be useful to have, a careful comparative analysis of the inter-pentameric interactions between echovirus 18 and other enteroviruses of known structures.

A: We have now included plots of the distribution of (1) charges and (2) hydrophobic areas at the inter-pentamer contacts of capsids of native virions and activated particles that were structurally characterized to date (SFig. 6). We have also included comparison of buried surface areas of inter-pentamer interfaces (STable 2). The comparison does not provide any clear indication explaining putative different behaviors of enteroviruses during genome release. We have now included a short discussion of these analyses in the manuscript (lines 98-102):

“A comparison of the distribution of charges and hydrophobic areas at the inter-pentamer contacts (SFig. 6) and a comparison of the inter-pentamer buried surface areas (STable 2) of enteroviruses do not provide evidence why distinct enteroviruses should employ different genome release mechanisms.”

This reviewer has also a number of concerns related to different technical aspects

I am a bit surprised that in the micrograph shown in figure 1, containing: active particles, empty particles and also particles releasing the RNA, all the particles releasing RNA are on the same side of the micrograph. Taking into account that to study the particles releasing RNA, samples should be diluted 4 times, I am wondering if the gradient of vitreous ice in the grid hole can exert any force on the particles frozen in the central region, making them to collapse. To exclude this, it would be nice to know if the particles releasing RNA are localized preferentially in the center of the grid holes.

A: We have now included four example micrographs (SFig. 1) to demonstrate that genome releasing particles are randomly distributed in quantifoil holes.

Figure 2 shows the reconstructions calculated for each of the 3 proposed models. Authors also show central sections of each reconstruction. I wonder if the presence of RNA could be masking what is really happening in the capsid and what they are seeing really is a capsid with the RNA region removed. To clarify this point, it would be of great help to have the reference-free two

dimensional classes of the final 3D refinement with C1 symmetry for each of the three RNA releasing reconstructions.

A: We have now included example reference-free 2D classes of the final 3D refinement with C1 symmetry in Fig. 2g-i.

In the 3D classifications, as well as in the 3D refinement, authors are using the models (lacking one, two or three pentamers), obtained in the prior 3D classification as initial models for the next steps. This fact could be introducing a bias in the reconstructions, specially if these reconstructions have not enough representative views. To discard this possible bias, It would be good to see the angle distribution of the particles.

A: Plots showing angle distributions of particles contributing to each of the reconstructions of the open particles have now been included in SFig. 3.

Coordinates and cryo-EM maps should be deposited.

A: Deposition codes for cryo-EM reconstructions and the corresponding PDB structures have now been included in the manuscript (STable 1).

Reviewers' Comments:

Reviewer #1:

Remarks to the Author:

The authors have carefully considered the points raised by the reviewers of their original manuscript and have made thoughtful and pertinent changes to the resubmitted version. The observations reported in the manuscript introduce a new, interesting and potentially very important alternative explanation of how picornaviruses uncoat to deliver their genomes into a fresh host cell. The work presented does not, of itself, provide conclusive evidence that the novel particle forms that they describe are on the direct path to genome entry. However, their presence is undeniable and it is very important for progress in the field that their role in cell entry is proven or refuted.

Reviewer #2:

Remarks to the Author:

The revised version addresses all issues raised by this reviewer.
My recommendation is to accept the manuscript